# Prediction of testosterone deficiency using different screening indexes in adult American men: An NHANES cross-sectional study

Bo Zhang[1,2☙], Yi Gu[3☙], Yuanyuan Li[4☙], Xingliang Feng🔟[1,2*]

1 Department of Urology, The Third Affiliated Hospital of Soochow University, Changzhou, Jiangsu, China, 2 Department of Urology, The First People's Hospital of Changzhou, Changzhou, Jiangsu, China, 3 Department of Breast Surgery, The Third Affiliated Hospital of Soochow University, Changzhou, Jiangsu, China, 4 Department of General Surgery, The Third Affiliated Hospital of Soochow University, Changzhou, Jiangsu, China

☙These authors contributed equally to this work.
* drf120@126.com

## Abstract

### Background

Testosterone levels are closely associated with visceral obesity, insulin resistance, and lipid metabolism. The objective of this study was to investigate the associations among eight indicators related to visceral obesity, insulin resistance, lipid metabolism, and testosterone levels.

### Methods

The data were obtained from a cross-sectional survey based on National Health and Nutrition Examination Survey (NHANES) 2013–2016. Logistic and linear regression were employed to assess the associations between these inicators and testosterone levels. Simultaneously, the receiver operating characteristic (ROC) curve was utilized to evaluate their predictive capacity for testosterone deficiency (TD).

### Results

Data from a collective of 1514 individuals selected from NHANES were analyzed. After adjusting all potential confounders, a tight association was identified between these eight indexes and TD. The ROC curve analysis showed that the triglyceride-glucose waist-to-height ratio (TyG-WHtR)is the best predictor of testosterone deficiency (AUC: 0.7760, 95%CI: 0.7460–0.8060), with a cut-off value of 5.375. Further analyses indicated that participants with higher TyG-WHtR index exhibitrd a lower total testosterone level(β: −79.36,95%CI: −105.90, −52.82). Additionally, males in TyG-WHtR index tertile 3 had a higher risk of TD (OR: 6.61, 95%CI: 2.90,15.07), and lower total testosterone levels (β: −121.9, 95%CI: −186.82, −56.98). All the results remained stable in the subgroup analyses stratified by diabetes and hypertension.

**Data availability statement:** All relevant data are within the paper and its Supporting Information files.

**Funding:** The author(s) received no specific funding for this work.

**Competing interests:** The authors have declared that no competing interests exist.

## Conclusions

We found that these indexes are tightly associated with testosterone levels in U.S. adult men. Moreover, the TyG-WHtR index demonstrates the most effective predictive performance in the population. However, more well-designed studies are still needed to validate their association.

## Introduction

Testosterone plays numerous essential roles in male physiological processes, including reproductive function, sexual activity, metabolism, inflammation regulation, and brain function [1,2]. After the age of 30, testosterone levels in men tend to decline progressively as a part of normal aging [3]. This decline can be further accelerated in the presence of metabolic disorders, often resulting in serum testosterone levels below 300 ng/dL, accompanied by a constellation of related symptoms [4]. Low testosterone levels in adult men may lead to a broad spectrum of adverse health consequences, including sexual dysfunction, reduced libido, decreased muscle strength, impaired cognitive function, poor cardiovascular health, and mood disturbances [5,6]. These manifestations collectively define testosterone deficiency syndrome (TDs), or hypogonadism [7]. Approximately 30% of adult men aged 40–79 years suffer from TD [5]. Metabolic diseases—such as obesity, insulin resistance, and dyslipidemia—are now recognized as major risk factors for TD [8]. These conditions can disrupt hypothalamic–pituitary–gonadal (HPG) axis regulation and suppress testosterone production through mechanisms involving chronic low-grade inflammation, oxidative stress, increased aromatase activity, and altered leptin signaling [9]. These findings indicate the importance of recognizing metabolic contributions to testosterone decline. In addition, the identification of metabolic indicators for predicting the development of TD may enable early recognition and targeted intervention, thereby playing a crucial role in reducing the overall disease burden and mitigating related complications.

Numerous studies have confirmed the inverse associations between TD and insulin resistance (IR) [10]. Testosterone secretion from Leydig cells in the testis could be decreased by IR, and the reduced testosterone levels could in turn exacerbate IR [11,12]. On this basis, several reliable indicators of IR have been proposed to predict TD, including the triglyceride-glucose index (TyG), and homeostasis model assessment of insulin resistance (HOMA-IR) index [13]. The former was developed from fasting triglycerides and glucose, while the latter was based primarily on fasting glucose and insulin [14,15]. Adipose tissue may contribute to reduced circulating testosterone levels through increased aromatization of androgens to estrogens, thereby disrupting the hypothalamic–pituitary–gonadal (HPG) axis. Considering visceral obesity as a sign of low testosterone, some studies also used lipid accumulation products (LAP) to predict TD, which was reported to be a reliable indicator of visceral obesity based on triglycerides (TG) and waist circumference (WC) [16,17]. Moreover, Amato et al. also proposed a novel body fat index, named visceral adiposity index (VAI), as a reliable indicator of visceral obesity. It was developed based on WC, BMI,

TG, and high-density lipoprotein cholesterol (HDL-c) [18]. However, the measurement procedure of the HOMA-IR index is complicated and is affected by exogenous insulin ejection, which is quite common in DM patients. Although both the LAP and VAI indexes do not incorporate fasting glucose levels, this characteristic may enhance their utility, as it avoids the confounding effects of glycemic fluctuations and allows for a more stable estimation of visceral adiposity compared to traditional measures such as BMI, WC, and waist-to-height ratio (WHtR) [19]. Many studies have demonstrated that the TyG index could reflect various metabolic conditions for people of varying health statuses [20–23]. However, the TyG index is not superior to HOMA-IR, and LAP in predicting TD [24,25].

More recently, emerging evidence found that the TyG-related parameters might have a higher predictive performance than TyG alone, which were developed based on TyG and obesity indices, including TyG-WC, TyG-BMI, and TyG-WHtR [23,26]. However, only a few studies investigated the associations between TyG-related parameters and testosterone levels in adult men. The predictive ability of these screening indexes for testosterone deficiency remains unclear. Therefore, the present study was conducted to investigate the relationships between different screening indexes and total testosterone levels. The variations among these screening indexes in predicting TD were also investigated, with the goal of identifying a superior and more comprehensive indicator for TD, which could provide more evidence to better manage male reproductive and sexual health.

## Materials and methods

### Study Population

The NHANES is a continuous cross-sectional survey conducted nationally and managed by the National Center for Health Statistics at the U.S. Centers for Disease Control and Prevention. The survey commenced in 1999 and has been conducted biennially using a complex multistage probability sampling, with the objective of deriving a representative sample from the non-institutionalized U.S. population. Certainly, specific sample weights would be assigned to selected participants to comprehensively reflect the health and nutritional conditions of the whole population. The data included five different sections: Demographics, Dietary, Examination, Laboratory, and Questionnaires. The demographics, diets, and questionnaires sections were mainly administered and collected by trained interviewers in the participants' homes, while the physical and laboratory examinations were conducted in the mobile examination centers by trained medical professionals with standardized physical examinations and laboratory tests. The Institutional Ethics Review Board of the National Centre for Health Statistics reviewed and approved the NHANES study protocols (Continuation of Protocol #2011–17), and all participants signed a written informed consensus before being enrolled the study. The NHANES database is open to the public and therefore the additional ethical review from our institutional ethics for this study was exempt.

Our current study used data from two survey cycles (2013–2014, 2015–2016), enrolling a total of 20,146 participants. Participants included in this study were adult males aged 20 years and older. The age cut-off was selected based on the availability of complete biochemical and anthropometric data in NHANES from this age onward, as well as the increasing prevalence of metabolic disorders, including obesity and insulin resistance, among younger adults. Consequently, we excluded (a) female participants (10251), (b) participants <20 years old (4390), (c) participants without screening indexes (3465), (d) participants without testosterone values (509), and (e) participants under sex hormone medications, including testosterone, estrogen, or "other sex hormone" (17). Finally, a total of 1514 eligible participants were included in our study for further analyses. The specific selection flow chart is displayed in Fig 1.

### Exposure and outcome definitions

All physical examinations were conducted in the mobile examination center, necessitating participants to observe a fasting period of at least eight hours. Height and weight measurements were conducted according to established protocols. During the measurement of waist circumference (WC), participants were instructed to stand naturally with their legs spread approximately 25–30 cm apart, at the conclusion of a normal exhale. Utilizing an inelastic ruler with a minimum

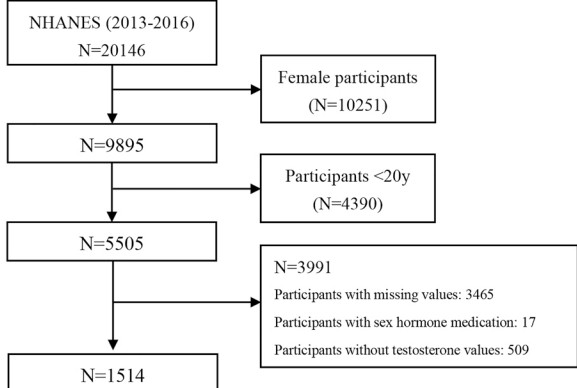

**Fig 1. Flow chart of the sample selection process.**

scale of one millimeter, WC was measured at the midpoint of the connecting line between the superior border of the iliac crest and the lower border of the 12th rib. Then, the ruler was horizontally ringed around the abdomen, and the reading was rounded to the nearest 0.1 cm.

The blood samples were collected in the morning (before 10.00 a.m.) after at least eight hours of fasting. The Roche Modular P and Roche Cobas 6000 chemistry analyzers were used to measure the triglycerides (TG), total cholesterol (TC), and HDL-cholesterol levels, while the hexokinase-mediated reaction on Roche/Hitachi Cobas C 501 chemistry analyzers were used to measure fasting glucose (FBG) levels. For insulin measurement, the AIA-PACK IRI was used.

All the screening indexes for the assessment of TD were calculated via the following formulas. The HOMA-IR was calculated and used as a reference index for assessing insulin resistance, given its established role and widespread use in epidemiological studies.

$$HOMA-IR \ = \ (\text{fasting glucose [mmol/L]} \ \times \ \text{fasting insulin [}\mu\text{U/ml]}/22.5)$$

$$LAP = \ [\text{WC (cm)} - 65] \ \times \ \text{TG (mmol/L) for males}$$

$$TyG \ = \ \text{Ln [TG (mg/dL)} \ \times \text{FPG (mg/dL)}/2]$$

$$VAI = \text{WC (cm)} \ / \ (39.68 \ + \ 1.88 \ \times \ \text{BMI (kg/m}^2)) \ \times \ \text{TG (mmol/L)}/ \ 1.03 \ \text{x} \ 1.31/\text{HDL}-\text{C (mmol/L) for males}$$

$$BMI = \text{weight (kg)}/\text{height}^2 \text{ (m)}$$

$$WHtR \ = \ \text{WC (cm)}/\text{height (cm)}$$

$$TyG-WC \ = \ TyG \ \times \ \text{WC (cm)}$$

$$TyG-BMI = \ TyG \ \times \ BMI$$

$$TyG-WHtR \ = \ TyG \ \times \ WHtR$$

The isotope dilution liquid chromatography-tandem mass spectrometry (ID-LC-MS/MS) method, based on the NIST reference method, was used to perform the measurements of the total amount of testosterone (https://wwwn.cdc.gov/Nchs/Nhanes/2013-2014/ TST_H.htm). The lower limit of detection (LLOD) of total testosterone was defined as 0.75 ng/ml, and the values below LLOD were replaced by the LLOD/sqrt (2). Detailed descriptions of the analyzers and methods are available on the NHANES website. In accordance with the American Urological Association guidelines, TD was defined as a total testosterone level below 300 ng/ml.

## Covariates

In the present study, TD was defined as the dependent variable, while the aforementioned screening indexes were defined as independent variables. Consequently, all the covariates were ranked based on previously published studies investigating TD in men, which indicated that these variables could potentially influence serum testosterone levels. The demographic characteristics defined as covariates included age, race, educational level, marital status, and poverty income ratio (PIR). In addition to the laboratory data, the LDL-c and uric acid (UA)levels were also collected for calculation. For the dietary data, the total energy, total fat, and total protein of the two-day interview were also included in our analysis. Alcohol consumption and smoking status were collected from the health questionnaires. Alcohol consumption was categorized as either yes or no, while smoking status was classified as never, former, or current. Furthermore, clinical diseases including hypertension and diabetes were recorded. Hypertension was defined as either a diastolic blood pressure $\geq 90$ mmHg, a systolic blood pressure $\geq 140$ mmHg, self-reported diagnosis of hypertension, or the use of antihypertensive medications. Diabetes was diagnosed if participants had fasting blood glucose levels exceeding 7.1 mmol/dL, or an oral glucose tolerance test $> 11.1$ mmol/L, or self-reported diabetes diagnosis, or HbAlc $\geq 6.5\%$, or used oral hypoglycemic agents or insulin.

## Statistical analysis

All statistical analyses adhered strictly to the CDC guidelines for NHANES statistical analyses. The appropriate sample weight was applied to all statistical analyses to account for unequal probabilities of NHANES participant selection, as well as nonresponse of those eligible and approached. The new sample weight for the combined survey cycles was calculated by dividing the 2-year weights for each cycle by 2. The basic characteristics were described based on the presence or absence of TD in the examination populations. Normality of continuous variables was assessed using the Shapiro–Wilk test. Variables with non-normal distribution were log-transformed prior to regression analyses. The continuous variables were presented as weighted survey mean and standard deviation (SD), while the categorical variables were shown as weighted survey means and 95 percent confidence intervals (95%CI). The survey-weighted linear regression (for continuous variables) and survey-weighted chi-square tests (for categorical variables) were employed to assess the difference between TD group and non-TD group.

The weighted regression models were employed to examine the associations between testosterone level and TD risk and various screening indexes (LAP, TyG, VAI, WC, BMI, TyG_WC, TyG_BMI, and TyG_WHtR). Specifically, linear regression models were used when serum total testosterone was treated as a continuous variable, while logistic regression models were applied when TD (TD, defined as TT<300 ng/dL) was analyzed as a binary outcome. And the corresponding results were reported as as β and 95%CI or odds ratios (ORs) and 95%CI. Model 1 consisted exclusively of independent variables; Model 2 was adjusted for age, race, educational level, marital status, and PIR; and Model 3 underwent further adjustments for BMI, hypertension, DM, smoking status, alcohol consumption, UA and LDL-C. Furthermore, interaction terms were incorporated to assess heterogeneity among different groups, with significance considered when P<0.05.

To assess the predictive performance of the screening indexes for TD, receiver operating characteristic (ROC) curves were plotted, and the areas under the curve (AUC) were compared. Furthermore, the ROC and AUC were performed on participants subgrouped based on hypertension and DM, aiming to to evaluate the robustness of the associations

between certain metabolic indexes (particularly TyG_WHtR) and TD among different sub-populations. The cut-off values for these screening indexes were established based on the sum of sensitivity and specificity. Additionally, the positive predictive value (PPV) and negative predictive value (NPV) were calculated.A generalized additive model regression and smoothed curve fitting were used to further explore the association between TD and the best predictive index. In this study, $p < 0.05$ was considered statistically significant. All of our analyses were performed using R version 4.0.5 and Empower-Stats software (www.empowerstats.com).

## Results

### Basic characteristics of eligible participants

In total, 1514 male participants were enrolled in our study for further analysis, and the detailed selecting process is shown in Fig 1. The detailed basic characteristics of eligible participants are displayed in Table 1. The mean age of the individuals was 48.34±0.70, with a prevalence of TD of 23.59%. No statistical differences in demographic characteristics were found between participants with and without TD, except for age and BMI. The mean total testosterone in the TD group was 224.59±5.15 ng/dl, while the mean total testosterone in the non-TD group was 514.68±6.67. No statistical differences were found in the dietary variables between participants with and without TD. Compared to individuals without TD, participants with TD showed a higher prevalence of DM and hypertension. Furthermore, participants with TD were more likely to be old and obese. All participants with TD showed higher indexes compared to participants without TD, including HOMA-IR, LAP, TyG, VAI, WC, BMI, TyG_WC, TyG_BMI, and TyG_WHtR.

### Associations between nine screening indexes and TD

The logistic regression results are displayed in Table 2. The positive associations between the screening indexes and TD were found in all three models. For model 1, TyG_WHtR showed the highest OR and 95%CI (OR: 3.24, 95%CI: 2.57,4.09, $p < 0.0001$), while TyG_WC presented the lowest OR and 95%CI (OR: 1.01, 95%CI: 1.01,1.01, $p < 0.0001$). When adjusted for age, race, educational level, marital status, and PIR, TyG_WHtR still presented the highest OR and 95%CI (OR: 3.23, 95%CI: 2.56,4.07, $p < 0.0001$). Similar results were achieved when adjusted for all covariates, with TyG_WHtR showing an OR and 95%CI of 3.30 (2.08,5.22), $p = 0.0001$. In addition, linear regression was performed to explore the association between the screening indexes and total testosterone, as displayed in Table 2. After adjusting for all covariates, the TyG_WHtR presented the highest β and 95%CI (β: −88.67, 95%CI: −101.53, −75.81).

### Predictive performance of nine screening indexes for predicting TD

Considering the important role of HOMA-IR in predicting TD, a total of nine screening indexes were enrolled to predict TD in American adults, including HOMA-IR, LAP, TyG, VAI, WC, BMI, TyG_WC, TyG_BMI, and TyG_WHtR. Overall, TyG_WHtR presented the highest AUC for TD (AUC: 0.7760, 95%CI: 0.7460–0.8060), while VAI showed the lowest AUC for TD (AUC: 0.6658, 95%CI: 0.6315–0.7002). Furthermore, TyG (AUC: 0.6752, 95%CI: 0.6397–0.7107) also showed bad performance for predicting TD compared to LAP (AUC: 0.7374, 95%CI: 0.7059–0.7689) and HOMA-IR (AUC: 0.7380, 95%CI: 0.7048–0.7711). The subgroup ROC analysis revealed stable results, with TyG_WHtR still presenting the highest AUC for TD, except for the participants with hypertension. The optimal cut-off value of TyG_WHtR to predict TD in the general population was 5.38, which decreased in the population without DM (4.98). The results are shown in Table 3 and Fig 2.

### Association between TyG_WHtR and TD in the overall population and subgroups

In the ROC analysis, TyG_WHtR presented the highest predictive ability for TD. Therefore, further analyses were conducted to verify the association between TyG_WHtR and testosterone levels with all the covariables adjusted (Table 4). When considering TyG_WHtR as a continuous variable, a strong positive association was observed between TyG_WHtR

**Table 1. Baseline characteristics of participants from 2013–2016 NHANES, weighted.**

| Characteristics | Total | TD(N = 289) | Non-TD(N = 1225) | p-Value |
|---|---|---|---|---|
| Age, years | 48.34 ± 0.70 | 53.37 ± 1.02 | 47.76 ± 0.76 | <0.0001 |
| LDL-cholesterol (mg/dL) | 113.15 ± 1.58 | 110.16 ± 2.98 | 113.80 ± 1.65 | 0.23 |
| HDL-cholesterol (mg/dL) | 49.32 ± 0.61 | 43.77 ± 0.83 | 50.54 ± 0.69 | <0.0001 |
| Uric acid (mg/dL) | 6.12 ± 0.05 | 6.54 ± 0.08 | 6.03 ± 0.05 | <0.0001 |
| Energy(kcal) | 2365.83 ± 32.92 | 2321.36 ± 80.01 | 2375.57 ± 35.25 | 0.53 |
| Total fat (gm) | 92.45 ± 1.68 | 91.87 ± 3.98 | 92.58 ± 1.61 | 0.85 |
| Total protein (gm) | 94.55 ± 1.46 | 93.96 ± 3.15 | 94.68 ± 1.47 | 0.82 |
| Total Testosterone (ng/dl) | 462.55 ± 8.80 | 224.59 ± 5.15 | 514.68 ± 6.67 | <0.0001 |
| HOMA-IR | 3.90 ± 0.24 | 6.90 ± 0.49 | 3.24 ± 0.29 | <0.0001 |
| LAP | 52.98 ± 2.25 | 87.20 ± 4.04 | 45.48 ± 1.86 | <0.0001 |
| TyG | 8.59 ± 0.03 | 8.96 ± 0.05 | 8.51 ± 0.03 | <0.0001 |
| VAI | 1.71 ± 0.06 | 2.41 ± 0.14 | 1.56 ± 0.06 | <0.0001 |
| WC | 102.16 ± 0.97 | 116.37 ± 1.96 | 99.05 ± 0.76 | <0.0001 |
| WHtR | 0.58 ± 0.01 | 0.66 ± 0.01 | 0.56 ± 0.00 | <0.0001 |
| TyG_WC | 881.56 ± 10.66 | 1044.28 ± 18.38 | 845.91 ± 8.33 | <0.0001 |
| TyG_BMI | 248.79 ± 3.77 | 307.10 ± 7.39 | 236.01 ± 2.85 | <0.0001 |
| TyG_WHtR | 5.03 ± 0.06 | 5.94 ± 0.10 | 4.82 ± 0.05 | <0.0001 |
| **Race** | | | | 0.68 |
| Black | 8.67(6.82,10.53) | 6.89(3.53,10.24) | 9.07(6.85,11.28) | |
| Mexican | 8.01(5.88,10.14) | 7.47(3.38,11.55) | 8.13(5.46,10.80) | |
| White | 68.34(55.56,81.11) | 70.95(63.10,78.79) | 67.77(62.64,72.90) | |
| Other | | | | |
| **Education** | 14.98(11.99,17.96) | 14.70(9.06,20.35) | 15.04(11.57,18.51) | 0.74 |
| Less than high school | 13.84(10.72,16.96) | 14.63(10.99,18.27) | 13.66(9.32,18.01) | |
| High school | 22.30(17.53,27.07) | 20.37(14.72,26.02) | 22.72(18.95,26.49) | |
| More than high school | 63.86(53.44,74.29) | 65.00(58.60,71.39) | 63.62(57.64,69.59) | |
| **Marital** | | | | 0.10 |
| Cohabitation | 68.68(58.00,79.35) | 75.19(66.74,83.64) | 67.25(62.89,71.61) | |
| Solitude | 31.32(26.85,35.80) | 24.81(16.36,33.26) | 32.75(28.39,37.11) | |
| **BMI** | | | | <0.0001 |
| <25 kg/m$^2$ | 28.42(23.76,33.08) | 9.23(3.52,14.94) | 32.62(28.29,36.95) | |
| 25-30 kg/m$^2$ | 37.57(32.82,42.32) | 26.83(20.75,32.92) | 39.92(35.71,44.13) | |
| ≥30 kg/m$^2$ | 34.01(26.27,41.76) | 63.94(54.91,72.96) | 27.46(22.89,32.03) | |
| **PIR** | | | | 0.81 |
| <1 | 11.89(9.44, 14.33) | 12.33(8.54,16.12) | 11.79(8.43,15.15) | |
| ≥1 | 88.11(75.62,100.61) | 87.67(83.88,91.46) | 88.21(84.85,91.57) | |
| **Hypertension** | | | | 0.005 |
| No | 58.14(50.40,65.89) | 44.90(36.06,53.73) | 61.05(57.14,64.95) | |
| Yes | 41.86(35.89,47.82) | 55.10(46.27,63.94) | 38.95(35.05,42.86) | |
| **Diabetes** | | | | <0.0001 |
| No | 82.88(72.30,93.47) | 69.03(61.88,76.18) | 85.92(83.84,88.00) | |
| Yes | 17.12(14.38,19.85) | 30.97(23.82,38.12) | 14.08(12.00,16.16) | |
| **Alcohol** | | | | 0.85 |
| No | 8.57(4.66, 12.48) | 8.06(0.97,15.14) | 8.69(5.31,12.06) | |
| Yes | 91.43(80.58,102.28) | 91.94(84.86,99.03) | 91.31(87.94,94.69) | |
| **Smoke** | | | | <0.0001 |

*(Continued)*

**Table 1.** (Continued)

| Characteristics | Total | TD(N = 289) | Non-TD(N = 1225) | p-Value |
|---|---|---|---|---|
| Never | 47.20(39.28,55.13) | 39.33(30.08,48.58) | 48.93(44.51,53.34) | |
| Former | 32.93(26.71,39.15) | 50.58(39.91,61.26) | 29.07(25.78,32.35) | |
| Now | 19.86(17.13,22.60) | 10.09(5.95,14.23) | 22.01(18.05,25.97) | |

TD: Testosterone Deficiency; HOMA-IR: Homeostasis Model Assessment of Insulin Resistance;

TyG: Triglyceride-Glucose index; LAP: Lipid Accumulation Products; VAI: Visceral Adiposity Index;

WC: Waist Circumference; WHtR: Waist-to-Height Ratio; BMI: Body Mass Index;

PIR: Ratio of Family Income to Poverty.

**Table 2.** The associations between screening indexes and TD and total testosterone, weighted.

| Variables | Model 1 | p-Value | Model 2 | p-Value | Model 3 | p-Value |
|---|---|---|---|---|---|---|
| TD-OR (95%CI) | | | | | | |
| LAP | 1.02(1.02,1.02) | <0.0001 | 1.02(1.02,1.02) | <0.0001 | 1.01(1.01,1.02) | 0.0002 |
| TyG | 2.75(2.16,3.49) | <0.0001 | 2.68(2.08,3.45) | <0.0001 | 1.89(1.23, 2.91) | 0.0075 |
| VAI | 1.42(1.27,1.59) | <0.0001 | 1.43(1.27,1.62) | <0.0001 | 1.23(1.03, 1.47) | 0.0235 |
| WC | 1.07(1.05,1.08) | <0.0001 | 1.07(1.05,1.08) | <0.0001 | 1.06(1.04,1.09) | 0.0002 |
| BMI | 1.16(1.12,1.20) | <0.0001 | 1.18(1.14,1.22) | <0.0001 | 1.17(1.09,1.25) | 0.0003 |
| TyG_WC | 1.01(1.01,1.01) | <0.0001 | 1.01(1.01,1.01) | <0.0001 | 1.01(1.00,1.01) | <0.0001 |
| TyG_BMI | 1.02(1.01,1.02) | <0.0001 | 1.02(1.01,1.02) | <0.0001 | 1.02(1.01,1.03) | <0.0001 |
| TyG_WHtR | 3.24(2.57,4.09) | <0.0001 | 3.23(2.56,4.07) | <0.0001 | 3.30(2.08,5.22) | 0.0001 |
| Total testosterone-$\beta$ (95%CI) | | | | | | |
| LAP | −1.58(−2.00, −1.16) | <0.0001 | −1.5(−1.94, −1.07) | <0.0001 | −0.73(−1.38, −0.08) | 0.0305 |
| TyG | −82.3(−103.53, −61.06) | <0.0001 | −75.47(−97.95, −52.99) | <0.0001 | −35.17(−72.59,2.25) | 0.0629 |
| VAI | −31.02(−45.77, −16.28) | <0.0001 | −29.12(−44.36, −13.89) | <0.0001 | −12.89(−30.50, 4.71) | 0.1353 |
| WC | −5.26(−5.91, −4.61) | <0.0001 | −5.15(−5.87, −4.44) | <0.0001 | −4.62(−6.21, −3.03) | <0.0001 |
| BMI | −12.74(−14.84, −10.64) | <0.0001 | −12.6(−14.69, −10.50) | <0.0001 | −10.76(−15.55, −5.98) | <0.0001 |
| TyG_WC | −0.5(−0.57, −0.43) | <0.0001 | −0.48(−0.56, −0.40) | <0.0001 | −0.44(−0.60, −0.27) | <0.0001 |
| TyG_BMI | −1.33(−1.53, −1.12) | <0.0001 | −1.29(−1.50, −1.08) | <0.0001 | −1.16(−1.58, −0.74) | <0.0001 |
| TyG_WHtR | −88.67(−101.53, −75.81) | <0.0001 | −86.67(−100.99, −72.35) | <0.0001 | −79.36(−105.90, −52.82) | <0.0001 |

Notes: Testosterone Deficiency; HOMA-IR: Homeostasis Model Assessment of Insulin Resistance; TyG: Triglyceride-Glucose index; LAP: Lipid Accumulation Products; VAI: Visceral Adiposity Index; WC: Waist Circumference; WHtR: Waist-to-Height Ratio; BMI: Body Mass Index; OR: odds ratio; CI, confidence interval. β, effect size for linear regression.

Model 1: Adjusted for none;

Model 2: Adjusted for age, race, educational level, marital status, and PIR

Model 3: Model 2, and furthermore adjusted for BMI, hypertension, DM, smoking status, alcohol consumption, and UA and LDL-c levels.

and TD. For the overall population, an OR of 3.30 was obtained, with a 95%CI between 2.08 and 5.22. When calculating the results in different subgroups, the results remained stable. However, the results were more pronounced in participants with DM (OR:4.086, 95%CI:2.488,6.711), and participants with hypertension (OR:3.645, 95%CI:1.522,8.561), compared to those without DM (OR:3.337, 95%CI:1.914,5.819) and without hypertension (OR:2.881, 95%CI:1.550,5.356). The regression results remained stable when the TyG_WHtR index was grouped into tertiles. Compared to participants in the lowest tertile, the participants in TyG_WHtR index tertile 3 displayed a higher risk of TD (OR:6.61, 95%CI:2.19,15.07). Similar results were obtained in the DM and hypertension sub-population, revealing a higher risk of TD in participants in

**Table 3. Selected parameters for predicting TD and the corresponding AUC, optimal cut-off values, their sensitivity and specificity, PPV and NPV.**

| TD | Variables | AUC (95%CI) | Cut-off values | Specificity (%) | Sensitivity (%) | PPV | NPV |
|---|---|---|---|---|---|---|---|
| **TD-overall** | | | | | | | |
| | HOMA-IR | 0.7380 (0.7048–0.7711) | 3.4093 | 0.7388 | 0.6298 | 0.3625 | 0.8943 |
| | LAP | 0.7374 (0.7059–0.7689) | 52.2989 | 0.6996 | 0.6644 | 0.3429 | 0.8983 |
| | TyG | 0.6752 (0.6397–0.7107) | 8.7545 | 0.6620 | 0.6298 | 0.3054 | 0.8834 |
| | VAI | 0.6658 (0.6315–0.7002) | 1.3665 | 0.5918 | 0.6574 | 0.2754 | 0.8799 |
| | WC | 0.7552 (0.7249–0.7854) | 102.4500 | 0.6261 | 0.7647 | 0.3255 | 0.9186 |
| | BMI | 0.7276 (0.6950–0.7602) | 28.9500 | 0.6482 | 0.6817 | 0.3137 | 0.8962 |
| | TyG_WC | 0.7713 (0.7411–0.8015) | 903.0824 | 0.6727 | 0.7509 | 0.3511 | 0.9196 |
| | TyG_BMI | 0.7519 (0.7200–0.7839) | 258.2105 | 0.6980 | 0.6886 | 0.3497 | 0.9048 |
| | TyG_WHtR | 0.7760 (0.7460–0.8060) | 5.3753 | 0.7478 | 0.6886 | 0.3917 | 0.9105 |
| **TD-participants without hypertension** | | | | | | | |
| | HOMA-IR | 0.7332 (0.6878–0.7786) | 4.4101 | 0.7793 | 0.6012 | 0.4689 | 0.8578 |
| | LAP | 0.7172 (0.6734–0.7610) | 63.3392 | 0.7316 | 0.6135 | 0.4255 | 0.8538 |
| | TyG | 0.6721 (0.6243–0.7200) | 8.7545 | 0.5924 | 0.6933 | 0.3553 | 0.8563 |
| | VAI | 0.6559 (0.6082–0.7035) | 1.9779 | 0.7396 | 0.4908 | 0.3719 | 0.8176 |
| | WC | 0.7187 (0.6735–0.7640) | 111.4000 | 0.7376 | 0.5951 | 0.4236 | 0.8490 |
| | BMI | 0.6979 (0.6505–0.7452) | 32.4500 | 0.7932 | 0.5215 | 0.4497 | 0.8365 |
| | TyG_WC | 0.7462 (0.7029–0.7895) | 958.2887 | 0.6759 | 0.7055 | 0.4137 | 0.8763 |
| | TyG_BMI | 0.7289 (0.6835–0.7742) | 260.3780 | 0.6342 | 0.7301 | 0.3927 | 0.8788 |
| | TyG_WHtR | 0.7574 (0.7151–0.7998) | 5.3753 | 0.6461 | 0.7669 | 0.4125 | 0.8953 |
| **TD-participants with hypertension** | | | | | | | |
| | HOMA-IR | 0.7204 (0.6697–0.7711) | 3.4093 | 0.7922 | 0.5714 | 0.3243 | 0.9137 |
| | LAP | 0.7350 (0.6871–0.7829) | 52.4412 | 0.7535 | 0.6270 | 0.3074 | 0.9205 |
| | TyG | 0.6552 (0.6004–0.7101) | 8.7199 | 0.6870 | 0.5794 | 0.2442 | 0.9035 |
| | VAI | 0.6630 (0.6110–0.7149) | 1.3266 | 0.6053 | 0.6587 | 0.2255 | 0.9104 |
| | WC | 0.7699 (0.7268–0.8130) | 100.6500 | 0.6773 | 0.7698 | 0.2939 | 0.9440 |
| | BMI | 0.7372 (0.6901–0.7842) | 30.0500 | 0.7812 | 0.5556 | 0.3070 | 0.9097 |
| | TyG_WC | 0.7736 (0.7288–0. 8184) | 903.0824 | 0.7632 | 0.6667 | 0.3294 | 0.9292 |
| | TyG_BMI | 0.7537 (0.7063–0.8011) | 288.8574 | 0.7396 | 0.6349 | 0.2985 | 0.9207 |
| | TyG_WHtR | 0.7726 (0.7275–0.8176) | 5.4056 | 0.8283 | 0.5873 | 0.3737 | 0.9200 |
| **TD-participants without DM** | | | | | | | |
| | HOMA-IR | 0.7010 (0.6575–0.7445) | 2.7378 | 0.6826 | 0.6364 | 0.2482 | 0.9194 |
| | LAP | 0.7118 (0.6694–0.7542) | 63.4026 | 0.8144 | 0.5152 | 0.3137 | 0.9107 |
| | TyG | 0.6203 (0.5738–0.6668) | 8.6166 | 0.6367 | 0.5758 | 0.2070 | 0.9011 |
| | VAI | 0.6368 (0.5911–0.6826) | 1.3576 | 0.6198 | 0.6061 | 0.2079 | 0.9052 |
| | WC | 0.7423 (0.7018–0.7828) | 101.5500 | 0.6567 | 0.7212 | 0.2570 | 0.9347 |
| | BMI | 0.7148 (0.6718–0.7578) | 30.1500 | 0.7525 | 0.5515 | 0.2684 | 0.9106 |
| | TyG_WC | 0.7423 (0.7014–0.7833) | 901.1600 | 0.7246 | 0.6545 | 0.2812 | 0.9272 |
| | TyG_BMI | 0.7252 (0.6822–0.7681) | 258.2105 | 0.7395 | 0.5939 | 0.2730 | 0.9171 |
| | TyG_WHtR | 0.7461 (0.7056–0.7867) | 4.9795 | 0.6337 | 0.7455 | 0.2510 | 0.9380 |
| **TD-participants with DM** | | | | | | | |
| | HOMA-IR | 0.7212 (0.6635–0.7790) | 4.7774 | 0.6682 | 0.7016 | 0.5404 | 0.8011 |
| | LAP | 0.7175 (0.6623–0.7727) | 52.0530 | 0.5650 | 0.7742 | 0.4974 | 0.8182 |
| | TyG | 0.6670 (0.6073–0.7286) | 9.1103 | 0.6726 | 0.6290 | 0.5166 | 0.7653 |

*(Continued)*

**Table 3.** (Continued)

| TD | Variables | AUC (95%CI) | Cut-off values | Specificity (%) | Sensitivity (%) | PPV | NPV |
|---|---|---|---|---|---|---|---|
| | VAI | 0.6605 (0.6013–0.7197) | 1.5313 | 0.5471 | 0.6774 | 0.4541 | 0.7531 |
| | WC | 0.6999 (0.6433–0.7565) | 113.7000 | 0.7578 | 0.5484 | 0.5574 | 0.7511 |
| | BMI | 0.6943 (0.6359–0.7526) | 31.2500 | 0.7220 | 0.6048 | 0.5474 | 0.7667 |
| | TyG_WC | 0.7361 (0.6818–0.7903) | 1009.6636 | 0.7130 | 0.6774 | 0.5676 | 0.7990 |
| | TyG_BMI | 0.7252 (0.6690–0.7814) | 279.9475 | 0.6996 | 0.6774 | 0.5563 | 0.7959 |
| | TyG_WHtR | 0.7464 (0.6924–0.8004) | 5.7621 | 0.6816 | 0.7177 | 0.5563 | 0.8128 |

Notes: Testosterone Deficiency; HOMA-IR: Homeostasis Model Assessment of Insulin Resistance; TyG: Triglyceride-Glucose index; LAP: Lipid Accumulation Products; VAI: Visceral Adiposity Index; WC: Waist Circumference; WHtR: Waist-to-Height Ratio; BMI: Body Mass Index; DM: Diabetes mellitus; AUC: Area under curve; OR: odds ratio.

TyG_WHtR index tertile 3 with DM and hypertension compared to the lowest tertile participants without DM and hypertension (OR:9.830, 95%CI:2.317,41.709; OR:10.417,95%CI:2.259,48.025). All the p-values for the trend were less than 0.001. The interaction terms indicated no dependence on DM and hypertension in these associations (DM: 0.454, Hypertension: 0.066). Additionally, the nonlinear relationship between the TyG_WHtR index and TD was characterized by smooth curve fitting, revealing a generally stable and linear association both in the general population and sub-populations (Fig 3).

The same analyses were also performed to explore the association between the TyG_WHtR index and total testosterone. The negative associations between the TyG_WHtR index and total testosterone were found in the fully adjusted model (β: −79.36, 95%CI: −105.90, −52.82). Similarly, the β was higher in participants with DM and hypertension (β: −88.936, 95%CI−136.526, −41.345; β: −89.331, 95%CI: −124.043, −54.620) compared to those without DM and hypertension. Moreover, the relationship remained statistically significant in the fully adjusted model when the TyG_WHtR index was divided into tertiles. All the p-values for the trend were less than 0.001, and there was no dependence on DM and hypertension in these associations (DM: 0.133, Hypertension: 0.157).

## Discussion

Our study was based on two continuous NHANES cycles, 2013–2014 and 2015–2016, and explored the association between two types of indexes and testosterone levels in U.S. adult males. The former were indicators for visceral obesity, including LAP, VAI, WC, and BMI, while the latter were indicators for insulin resistance, including TyG, TyG-WC, TyG-BMI, and TyG-WHtR. Furthermore, their performance in predicting the risk of TD was assessed. All these indexes were associated with testosterone levels, with the TyG-WHtR index showing the best predictability for TD. Males with a higher TyG-WHtR index presented lower total testosterone levels and increased risk of TD; these associations remained stable and significant in males with or without DM and hypertension.

In a study enrolling 149 men aged 18–66 years with a mean BMI of 42.7 kg/m$^2$, the authors reported that about 35.6% of obese men exhibited abnormally low testosterone levels, which was inversely related to the BMI [27]. Obesity, especially visceral obesity, may indeed result in HPT dysfunction, and consequently secondary TD. BMI has been used to assess obesity-related sex hormone disorders [28]. However, BMI does not account for body fat distribution, and visceral adipose tissue is not detected by BMI. In contrast, WC is a traditional indicator that is easily standardized and clinically applicable [29]. A cross-sectional study enrolled community adult men and reported that WC was negatively related to testosterone levels; the associations appeared to be stronger and more consistent than BMI [30]. Furthermore, another cross-sectional study of children and adolescents also highlighted WC as a significant indicator of the effect of sex hormone levels [31]. Recently, several novel indexes, including VAI and LAP, were found to accurately and rapidly identify visceral obesity;

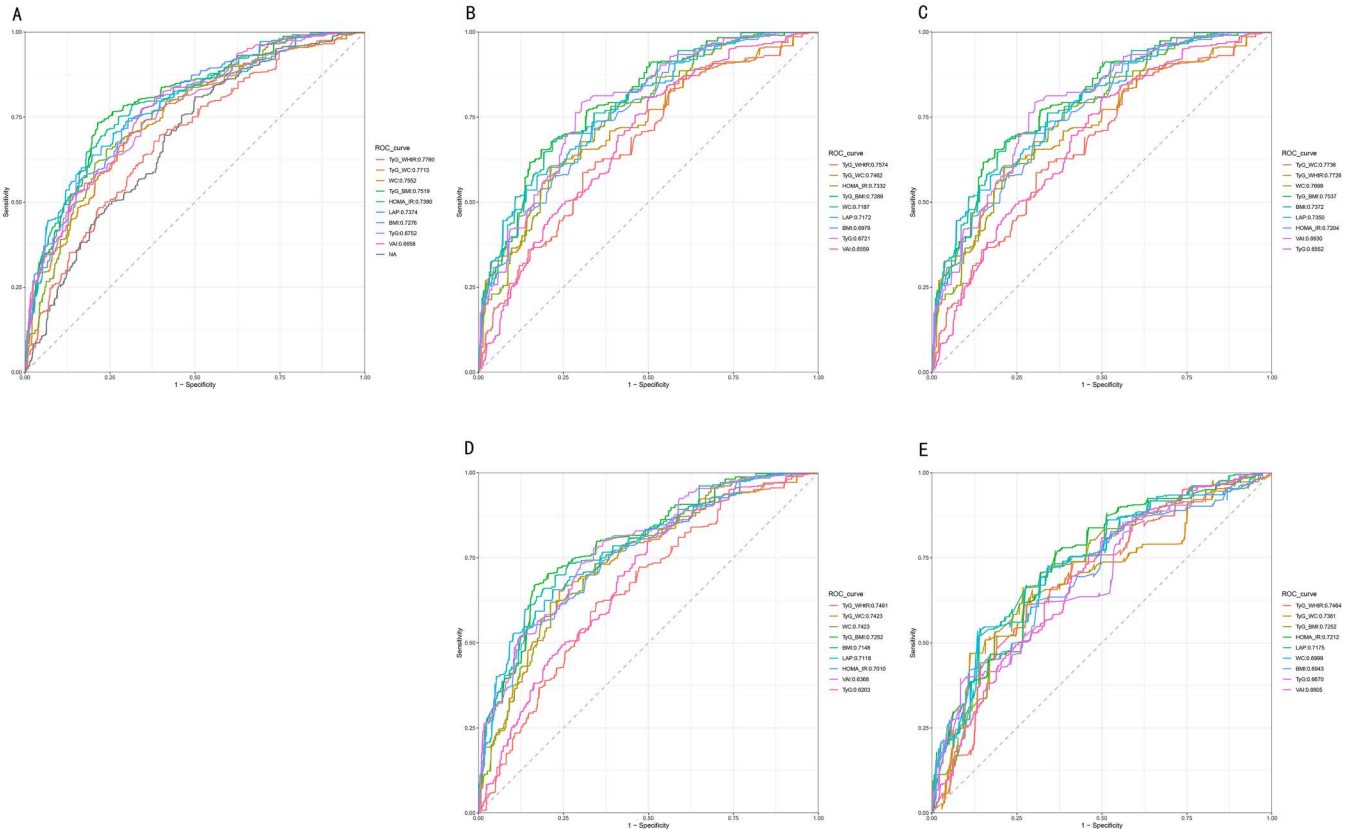

**Fig 2. Receiver operating characteristic curves of the eight indexes for identifying testosterone deficiency.** (A: Overall population; B/D: without or with Hypertension; C/E: without and with Diabetes).

these were superior to BMI, and WC, as they comprehensively included the laboratory index [32–34]. In a recent study, VAI and LAP were found to be strongly correlated with total testosterone levels in type 2 DM patients. The cut-off values for predicting TD were VAI > 3.985, and LAP > 51.654, respectively [35].

Estradiol and leptin are considered key molecules in the pathophysiological mechanisms of obesity and TD [36]. Adipose tissues secrete estradiol by converting testosterone using aromatase. Additionally, excessive estradiol could have deleterious effects on male testis function by suppressing the HPT axis [37,38]. Leptin, another vital hormone, is predominantly secreted by white adipocytes, and also plays a vital medicator on the HPT axis. Theoretically, leptin could stimulate the hypothalamus to secrete and release GnRH, thus regulating the HPT axis to secrete and release more testosterone in circulation [39]. However, obese individuals often exhibit leptin resistance, although higher circulatory leptin levels were found. Furthermore, excess leptin acts on Leydig cells to inhibit steroidogenesis [39].

Insulin resistance is also related to obesity and testosterone deficiency. A cross-sectional study published in 2008 demonstrated that low testosterone levels were independently associated with IR in men with type 1 or 2 DM [12]. A retrospective study enrolled obese men and indicated that lower testosterone levels were associated with high IR [40]. However, the underlying mechanisms remain unclear. IR promotes the secretion of estradiol, which could suppress the secretion of testosterone [10]. Furthermore, hyperinsulinism could induce DAX-1 expression in Leydig cells and inhibit steroidogenesis, ultimately resulting in reduced testosterone secretion [41]. Additionally, insulin action and insulin sensitivity in the brain are associated with the preservation of the functional integrity of the HPT axis [42]. An animal study showed

**Table 4. The association between TyG_WHtR and TD in overall population and stratified by hypertension and DM.**

| TyG_WHtR | Overall | Participants with hypertension | Participants without hypertension | Participants with DM | Participants without DM |
|---|---|---|---|---|---|
| TD-OR (95%CI) | | | | | |
| Continuous | 3.30 (2.08,5.22) | 3.645(1.552, 8.561) | 2.881(1.550,5.356) | 4.086(2.488, 6.711) | 3.337(1.914,5.819) |
| Tertile 1 | 1 (Ref) | 1 (Ref) | 1 (Ref) | 1 (Ref) | 1 (Ref) |
| Tertile 2 | 1.32(0.51, 3.43) | 0.914(0.224, 3.732) | 1.667(0.634, 4.382) | 0.999(0.219, 4.556) | 1.163(0.423, 3.198) |
| Tertile 3 | 6.61(2.90,15.07) | 10.417(2.259,48.025) | 4.804(1.282,18.003) | 9.830(2.317,41.709) | 5.395(1.995,14.589) |
| P for trend | <0.001 | <0.001 | <0.001 | <0.001 | <0.001 |
| P for interaction | | 0.066 | | 0.454 | |
| Total testosterone-$\beta$ (95%CI) | | | | | |
| Continuous | −79.36(−105.90, −52.82) | −89.331(−124.043, −54.620) | −73.353(−96.545, −50.161) | −88.936(−136.526, −41.345 | −70.632(−105.199, −36.065) |
| Tertile 1 | 1 (Ref) | 1 (Ref) | 1 (Ref) | 1 (Ref) | 1 (Ref) |
| Tertile 2 | −32.91(−75.74, 9.93) | −49.372(−135.733, 36.988) | −30.39(−69.307, 8.528) | 35.68(−56.017,127.377) | −29.062(−74.971, 16.847) |
| Tertile 3 | −121.9(−186.82, −56.98) | −184.385(−302.247, −66.523) | −92.441(−153.219, −31.664) | −145.788(−226.953, −64.624) | −100.363(−179.553, −21.174) |
| P for trend | <0.001 | <0.001 | <0.001 | <0.001 | <0.001 |
| P for interaction | | 0.157 | | 0.133 | |

Notes: Testosterone Deficiency; TyG: Triglyceride-Glucose index; WHtR: Waist-to-Height Ratio; DM: Diabetes mellitus. OR: odds ratio; CI, confidence interval. β, effect size for linear regression. Model 3: Model 2, and furtherly adjusted for BMI, hypertension, DM, smoking status, alcohol consumption, and UA and LDL-c levels.

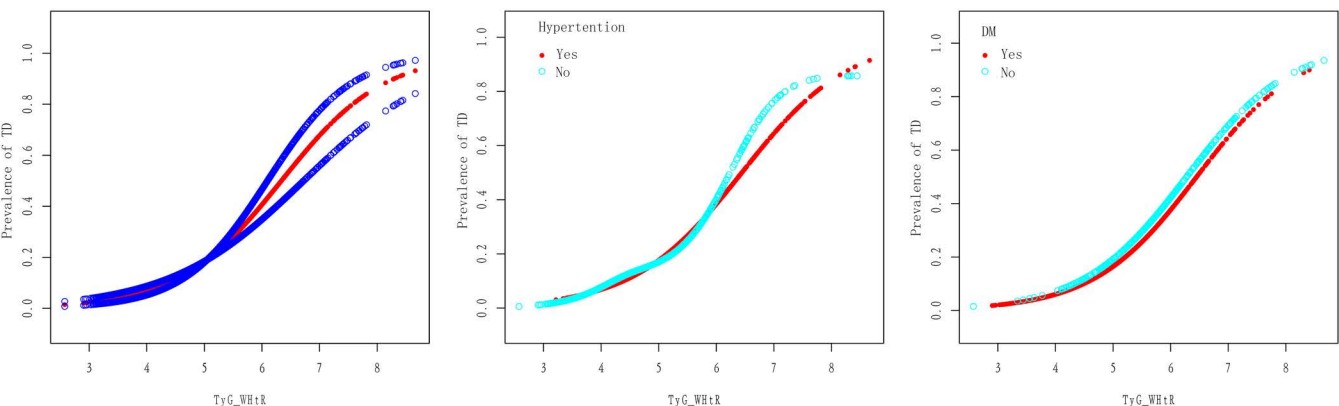

**Fig 3. Smoothed curve fit between the TyG_WHtR index and total testosterone in overall populations and participants with or without diabetes and hypertension.**

that LH and testosterone secretion are greatly reduced when the insulin receptor is selectively deleted from the neurons, and GnRH secretion from hypothalamic neurons is improved when incubated with insulin [42].

In addition to obesity and IR, dyslipidemia may also be associated with TD. A cohort study enrolling 384 patients reported that declined testosterone levels were related to the presence of dyslipidemia [43]. Another larger sample study enrolling 1619 middle-aged men also indicated that increased circulatory triglyceride levels were associated with low testosterone levels [44]. The mechanisms underlying the correlation between dyslipidemia and low testosterone levels

may involve proinflammatory substances (e.g., TNF-α, IL-8, IL-6, and IL1β). Dyslipidemia could stimulate endothelial cells and monocytes to secrete more inflammatory factors. IL-6 and TNF-α produced from interstitial macrophages could impair Leydig cell testosterone secretion and release by aggravating inflammation and reactive oxygen species [45]. Additionally, TNF-α could also inhibit steroidogenesis in Leydig cells at the transcriptional level of steroidogenic enzymes [46].

Recently published studies have compared the predictive performance of several indexes for TD, and further explored the tight associations between them, including HOMA-IR, LAP, and TyG. The theoretical basis of these indexes includes the aforementioned aspects, such as obesity, insulin resistance, and dyslipidemia. The studies indicated that HOMA-IR and LAP had a comparable predictive ability for TD, which was better than TyG. However, the TyG index was cheaper and more accessible than the measurement procedure of insulin for HOMA-IR. Furthermore, the TyG index not only considers glucose metabolism but also evaluates the lipid profile metabolism. Consequently, TyG is still relevant in predicting TD. Our study showed that the TyG-related indexes presented better predictive ability than HOMA-IR and LAP. Interestingly, TyG-WC and TyG-WHtR had a higher predictive performance for TD in the overall population, and in participants with or without DM and hypertension. In our study, the HOMA-IR index had an AUC of 0.7380, and the LAP index had an AUC of 0.7374, which were all comparable to the published results (HOMA-IR:0.71, LAP:0.719).

The association between the best predictive index, TyG-WHtR, testosterone decline, and TD was analyzed. Subgroup analysis based on hypertension and DM also showed similar results. In contrast, LAP showed a tighter association in participants with hypertension, and TyG showed a more significant association in participants with DM. A possible explanation may be that the TyG-WHtR was adjusted with visceral obesity indexes, and it could reflect the obesity, glucose, insulin resistance, and lipid profile metabolism. Interestingly, TyG-WHtR showed better predictive performance than TyG-WC. This may be attributed to the WHtR index being an adjusted index with WC and height, which could better reflect abdominal obesity than WC alone.

Actually, the TyG index was used extensively in many clinical diseases. A study found that the TyG index could have a better predictive performance than HMOA-IR for early-stage chronic kidney disease and metabolic syndrome [47]. For andrological diseases, the TyG index was associated with the prevalence of erectile dysfunction (ED) and could serve as a better predictor of ED (AUC = 0.739, sensitivity = 67%, specificity = 68.6%) [48]. Therefore, the role of TyG and TyG-related parameters in general populations should not be disregarded. Combining TyG and WHtR could improve their predictive ability for TD. Moreover, these indexes have been used in many clinical diseases, such as non-alcoholic fatty liver disease and metabolic dysfunction-associated fatty liver disease [49].

To our best knowledge, we are the first one to explore the association between these screening indexes and TD. This study further compares their predictive performance for TD using a nationally representative population. The present study had a number of strengths. Firstly, the representative study sample made the findings more generalizable. Secondly, detailed sociodemographic and clinical disease factors were included, which allowed for the adjustment of confounders and subsequent subgroup analysis. Nevertheless, the limitations of the current study should be acknowledged. Firstly, the temporal relationship between these indexes and testosterone cannot be determined due to the cross-sectional design of NHANES. Secondly, the use of recall questionnaire data, such as smoking and alcohol status, may lead to recall bias. Thirdly, we acknowledge that the diagnosis of TD in our study was based solely on a single morning measurement of total testosterone, as provided by the NHANES dataset. This approach, while common in epidemiological research, may overestimate the prevalence of hypogonadism since clinical guidelines recommend confirmation with at least two separate measurements. Fourth, although symptoms such as decreased energy, depression, reduced libido, and erectile dysfunction play a key role in the clinical evaluation of testosterone deficiency, they are non-specific and were not included in our analysis. We have clarified that the absence of symptom assessment is an important limitation, and that symptoms alone are insufficient for diagnosis without biochemical confirmation. Furthermore, participants aged< 20 years were excluded from our study, and more studies should be conducted enrolling wider populations.

## Conclusion

This study found that LAP, VAI, WC, and BMI and TyG, TyG-WC, TyG-BMI, and TyG-WHtR, are closely associated with testosterone levels in U.S. adult men. These parameters could be used to identify the risk of TD. Among these screening indexes, TyG-WHtR demonstrates the best predictive performance in the population, surpassing other indexes, especially considering IR or visceral obesity. Furthermore, the detailed analysis showed that men with higher TyG-WHtR tend to face an increased risk of TD, irrespective of DM and hypertension status. However, more well-designed studies are needed to validate the association between TyG-WHtR and testosterone level, and further explore the underlying mechanisms between them.

## Supporting information

**S1 File. Mininal data set for final analysis.**
(TXT)

**S2 File. Variables for mininal data set.**
(XLSX)

## Acknowledgments

Thanks to all NHANES participants and staff, and thanks to the Third Affiliated Hospital of Soochow University for their support.

## Author contributions

**Conceptualization:** Bo Zhang, Yi Gu, Xingliang Feng.

**Data curation:** Bo Zhang, Yi Gu, Yuanyuan Li, Xingliang Feng.

**Formal analysis:** Bo Zhang, Yi Gu, Yuanyuan Li, Xingliang Feng.

**Funding acquisition:** Yuanyuan Li, Xingliang Feng.

**Investigation:** Bo Zhang, Xingliang Feng.

**Methodology:** Bo Zhang, Yi Gu, Xingliang Feng.

**Project administration:** Xingliang Feng.

**Resources:** Yuanyuan Li, Xingliang Feng.

**Supervision:** Xingliang Feng.

**Validation:** Xingliang Feng.

**Writing – original draft:** Bo Zhang.

**Writing – review & editing:** Bo Zhang, Xingliang Feng.

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
