## [Decision Letter · Decision Letter 0]

21 Mar 2025

PONE-D-24-22453Prediction of testosterone deficiency using different screening indexes in adult American men: an NHANES cross-sectional studyPLOS ONE

Dear Dr. Feng,

Thank you for submitting your manuscript to PLOS ONE. After careful consideration, we feel that it has merit but does not fully meet PLOS ONE’s publication criteria as it currently stands. Therefore, we invite you to submit a revised version of the manuscript that addresses the points raised during the review process.

We look forward to receiving your revised manuscript.

Kind regards,

Shaonong Dang, PhD

Academic Editor

PLOS ONE

Journal Requirements:

“NO authors have competing interests”

Additional Editor Comments:

It is an interesting study in which authors invesigated association testosterone deficiency and different screening indexes. The reviewers have raised some comments, and the authors are suggested to address them carefully.

Reviewers' comments:

Reviewer's Responses to Questions

**Comments to the Author**

1. Is the manuscript technically sound, and do the data support the conclusions?

Reviewer #1: Yes

Reviewer #2: Yes

2. Has the statistical analysis been performed appropriately and rigorously? 

Reviewer #1: Yes

Reviewer #2: Yes

3. Have the authors made all data underlying the findings in their manuscript fully available?

Reviewer #1: Yes

Reviewer #2: Yes

4. Is the manuscript presented in an intelligible fashion and written in standard English?

Reviewer #1: Yes

Reviewer #2: Yes

5. Review Comments to the Author

Reviewer #1: Dear editor and authors

I am very pleased and honored to review a manuscript for PLOS ONE. The article entitled “Prediction of testosterone deficiency using different screening indexes in adult American men: an NHANES cross-sectional study” provides a cross-sectional survey from the National Health and Nutrition Examination Survey (NHANES) database aimed at evaluating the role of metabolic indexes in predicting testosterone levels and testosterone deficiency. The association between metabolic disease (e.g., Type 2 Diabetes, overweight/obesity, etc.) and testosterone deficiency is well established. Nevertheless, many patients might be asymptomatic or present very non-specific symptoms, making the diagnosis of testosterone deficiency in this clinical setting still challenging, and contributing to the misdiagnosis of this condition. Current literature data in this clinical context are scanty and the data reported in the study provides a very interesting insight, thus contributing to early diagnosis and treatment of hypogonadal patients.

After carefully reading the whole manuscript, my comments and suggestions are reported below in bulleted list.

1) TITLE and ABSTRACT

The title is clear and coherent with the text. The abstract is also well-structured: all sections are summarized in a clear, balanced, and comprehensive way.

I suggest to add in this paragraph the type of study design (cross-sectional survey), since it was reported only in the “Introduction” but not in this section of the manuscript.

2) INTRODUCTION

In the early section of this paragraph, I suggest to describe more in detail the relationship between testosterone decline/hypogonadism and metabolic diseases (considering both the pathophysiological and clinical implications). I also suggest to include in this overview a brief explanation of the “functional hypogonadism” that often characterize these subjects.

In this paragraph authors states that “Considering visceral obesity ad a symptom of low testosterone, …”. I suggest to replace the term SYMPTOM with SIGN, since obesity represents a constitutional feature.

The description of the LAP index might be misleading, since its independency from serum glucose level might be perceived by the reader as a limitation. However, the wide variability in glucose levels and the potential benefit of this index (as well as VAI) in estimating the visceral adiposity should be pointed out more clearly. From this perspective, being independent from serum glucose concentration might actually represent an advantage for these indexes.

At the end of the paragraph authors point out that data were derived from the NHANES database; I suggest consider to move this information into the next paragraph (Methods).

3) METHODS

Was this study approved by ethical committee? If so, I suggest to add the “approval number” or the “registration code”. All participants signed a written informed consensus before being enrolled the study.

When presenting the inclusion criteria, I suggest to briefly describe why you chose “20 years” as the cut-off for inclusion. On the one hand, due to the high prevalence of overweight in the American population, including such young patients might represent a strong point. On the other hand, including males older than 50 years might introduce a potential confounding factor, since in these subjects the testosterone reduction is at least in part cause by physiological reduction of testicular steroidogenesis over time, regardless from the presence of metabolic disease.

When describing the technique for sample collection, I suggest to also add the time of collection (e.g., “samples were collected in the morning (before 10.00 a.m.) after at least eight hours of fasting”).

When describing all indexes considered, I suggest to point out the role of HOMA-index to make it explicit is role (I suppose it considered as a “gold-standard reference”)

Finally, when describing cut-off for diagnosing diabetes, please correct a small typing error ("blood glucose levels" is expressed in mg/dL instead of mmol/L: it should be 7.1 mmol/L and not 7.1 mg/dL).

When considering specifically the “statistical analysis section”:

- I suggest to add in the text whether and how continuous variables were testes for normal distribution (e.g., Shapiro Wilk test) and if/how were they transformed if requested (was testosterone serum level normally distributed or was it necessary to transform it before regression analysis?);

- I suggest to rephrase when describing the types of regression used, in order to make it simpler for the reader to understand the logic underlying the statistic process (i.e., testosterone was considered both as continuous variable (➝ linear regression) and dichotomic variable (➝ logistic regression)).

- When reading the manuscript and tables I guess that multivariate analysis was performed rather than simple ones: from this perspective I suggest to make it explicit into the text and adding into tables (i.e., table 2) the statistical parameters of every model (R, R2, p (test F) coefficient for linear regression and Likelihood ratio, R2, adjusted R2 and p for logistic one).

- Finally, consider making explicit that also subgroup analyses were performed to test the reliability of the predictive potential of some index (i.e., TyG_WHtR).

4) RESULTS

Information provided at the beginning of the paragraph (patients ages > 20 years) should be removed, since it is already present in the “Method” section.

The first phrase of the section “screening index for predicting TD” should also be moved into the introduction. I suggest to start this paragraph by directly describing the main results of the study.

5) DISCUSSION and CONCLUSIONS

I consider to re-structure the whole paragraph and consider to move some information into the “Introduction”, as the description of hypogonadism (despite being interesting and well described, I suggest to eliminate the description of primary and secondary hypogonadism since it is out of the primary focus of the manuscript. I rather suggest adding few considerations about the so called “functional hypogonadism”).

I also suggest moving into the “Introduction” the description about metabolic indexes (BMI, WC, LAP and VAI) and testosterone deficiency, as well as the role of insulin-resistance (IR) and testosterone deficiency.

Authors also describe in detail the pathophysiological role of estrogen and leptin on pituitary-gonadal axis. Although being very interesting and well described, this information might be out of the main scope of the present manuscript. I suggest to resume this section in one phrase (into the “introduction”) highlighting the potential role of adipose tissue in reducing plasma testosterone due to the aromatization processes.

More in general, the “Discussion” should be enriched by pointing out the potential benefits of applying these indexes in clinical practice, as well as advantages in early diagnosis and treatment of hypogonadism.

When discussing the limitations of the study I suggest to consider the following aspects: 1) discuss that making diagnosis of testosterone deficiency based only on a single measurement may lead to overestimation of hypogonadism; 2) despite being very important in the clinical evaluation and treatment of these patients, the presence of symptoms (i.e., decreased energy, depression, reduced libido and erectile deficiency) does not allow per se to diagnose testosterone deficiency.

6) FIGURES

Data are displayed accurately, are easy to interpret and are consistent with data analysis presented in the manuscript.

All tables and figure present an accurate legend.

After addressing the above issues, I believe this manuscript is highly worthy of publication. I look forward to the authors resolving my concerns.

Kind regards.

Reviewer #2: The manuscript is technically sound and the data supported the conclusions. Statistical analysis is adequate and performed appropriately and rigorously. The authors have made all data underlying the findings in their manuscript fully available. The manuscript is presented in an intelligible fashion and written in standard English, though there are few spelling mistakes. For instance:

line 19: indicators not inicators

line 24: exhibited not exhibitrd

line 27: indexes not indexses

line54: injection not ejection

line 219: mediator not medicator

6. PLOS authors have the option to publish the peer review history of their article (what does this mean? ). If published, this will include your full peer review and any attached files.

**Do you want your identity to be public for this peer review?** For information about this choice, including consent withdrawal, please see our Privacy Policy .

Reviewer #1: No

Reviewer #2: **Yes: ** Professor Sikiru Adetona OLURODE

---

## [Author Response · Author response to Decision Letter 1]

25 Mar 2025

Dear Editor and Reviewers,

We are sincerely grateful for your letter and the positive evaluation of our manuscript. We were truly pleased to receive your encouraging comments and constructive suggestions. Although the review process took some time, we fully understand that the increased number of submissions, driven by the growing influence of the journal, has inevitably led to delays. We highly appreciate the time and effort the reviewers have devoted to reviewing our work. We have carefully considered every comment and discussed each suggestion thoroughly among all co-authors. Below, we provide point-by-point responses to all the reviewers’ comments, with detailed explanations of the revisions made.

Additional Editor Comments:

We sincerely thank the Editor for the positive feedback and for recognizing the value of our study. We greatly appreciate your summary of the reviewers’ comments. We have carefully addressed each point raised and revised the manuscript accordingly to improve its clarity, scientific rigor, and overall quality. A detailed point-by-point response to all reviewer comments is provided below.

Reviewer 1

Dear Reviewer 1,

We sincerely thank you for your positive evaluation of our manuscript and for giving us the opportunity to revise and improve our work. Your detailed and insightful comments reflect the time, effort, and professional expertise you devoted to reviewing our study. We truly appreciate your thoughtful suggestions, which have been immensely helpful in enhancing the quality of our manuscript. All the comments have been carefully discussed among the co-authors, and we have provided point-by-point responses and corresponding revisions to address each of your concerns to the best of our ability. We hope that our responses meet your expectations and demonstrate our commitment to improving the manuscript.

Q1) TITLE and ABSTRACT

>>>Q1.1 The title is clear and coherent with the text. The abstract is also well-structured: all sections are summarized in a clear, balanced, and comprehensive way.

>>>Answer: We sincerely thank the reviewer for the positive comments on our manuscript title and abstract. We are glad that you found them clear, coherent, and well-structured. Your recognition is greatly appreciated and encourages us in our research and writing.

>>>Q1.2 I suggest to add in this paragraph the type of study design (cross-sectional survey), since it was reported only in the “Introduction” but not in this section of the manuscript.

>>>Answer: Thank you for your valuable suggestion. As advised, we have added the description of the study design- “a cross-sectional survey” -in the relevant paragraph to ensure clarity and consistency throughout the manuscript. We agree that specifying the study design in this section helps enhance the transparency and scientific rigor of the methodology.

Q2) INTRODUCTION

>>>Q2.1 In the early section of this paragraph, I suggest to describe more in detail the relationship between testosterone decline/hypogonadism and metabolic diseases (considering both the pathophysiological and clinical implications). I also suggest to include in this overview a brief explanation of the “functional hypogonadism” that often characterize these subjects.

>>>Answer: We greatly appreciate your insightful comment. In response, we have substantially revised the corresponding section of the Introduction to provide a more comprehensive overview of the bidirectional relationship between testosterone deficiency and metabolic disorders. Specifically, we elaborated on both the pathophysiological mechanisms (e.g., chronic low-grade inflammation, oxidative stress, increased aromatase activity, and altered leptin signaling) and the clinical implications (e.g., increased risk of type 2 diabetes, cardiovascular disease, and mood or cognitive impairment). Furthermore, we included an explanation of “functional hypogonadism,” highlighting its reversible nature and its frequent association with metabolic conditions such as obesity and insulin resistance. These additions provide a clearer rationale for our study and better contextualize the relevance of identifying metabolic indicators for early detection and intervention in testosterone deficiency. The revised paragraph now reads as follows: "Testosterone plays numerous essential roles in male physiological processes, including reproductive function, sexual activity, metabolism, inflammation regulation, and brain function. After the age of 40, testosterone levels in men tend to decline progressively as a part of normal aging. This decline can be further accelerated in the presence of metabolic disorders, often resulting in serum testosterone levels below 300 ng/dL, accompanied by a constellation of related symptoms. Low testosterone levels in adult men may lead to a broad spectrum of adverse health consequences, including sexual dysfunction, reduced libido, decreased muscle strength, impaired cognitive function, poor cardiovascular health, and mood disturbances. These manifestations collectively define testosterone deficiency syndrome (TDs), or hypogonadism. Approximately 30% of adult men aged 40 to 79 years suffer from TD. Metabolic diseases—such as obesity, insulin resistance, and dyslipidemia—are now recognized as major risk factors for TD. These conditions can disrupt hypothalamic–pituitary–gonadal (HPG) axis regulation and suppress testosterone production through mechanisms involving chronic low-grade inflammation, oxidative stress, increased aromatase activity, and altered leptin signaling. These findings indicate the importance of recognizing metabolic contributions to testosterone decline. In addition, the identification of metabolic indicators for predicting the development of testosterone deficiency may enable early recognition and targeted intervention, thereby playing a crucial role in reducing the overall disease burden and mitigating related complications."

>>> Q2.2 In this paragraph authors states that “Considering visceral obesity ad a symptom of low testosterone, …”. I suggest to replace the term SYMPTOM with SIGN, since obesity represents a constitutional feature.

>>>Answer: Thank you for your careful observation. We agree with your comment that “sign” is more appropriate than “symptom” in this context, as visceral obesity is an objectively measurable physical finding rather than a subjective complaint. Accordingly, we have replaced the term “symptom” with “sign” in the revised manuscript.

>>> Q2.3 The description of the LAP index might be misleading, since its independency from serum glucose level might be perceived by the reader as a limitation. However, the wide variability in glucose levels and the potential benefit of this index (as well as VAI) in estimating the visceral adiposity should be pointed out more clearly. From this perspective, being independent from serum glucose concentration might actually represent an advantage for these indexes.

>>>Answer: Thank you very much for this thoughtful and constructive suggestion. We fully agree that the independence of the LAP and VAI indexes from serum glucose levels should not be viewed as a limitation. On the contrary, given the wide variability and daily fluctuations in glucose concentrations, the glucose-independent nature of LAP and VAI may enhance their stability and reliability in estimating visceral adiposity. To clarify this point, we have revised the corresponding section of the manuscript to emphasize this potential advantage, and to prevent any misinterpretation regarding the clinical utility of these indexes: “Although both the LAP and VAI indexes do not incorporate fasting glucose levels, this characteristic may enhance their utility, as it avoids the confounding effects of glycemic fluctuations and allows for a more stable estimation of visceral adiposity compared to traditional measures such as BMI, WC, and waist-to-height ratio (WHtR).”

>>> Q2.4 At the end of the paragraph authors point out that data were derived from the NHANES database; I suggest consider to move this information into the next paragraph (Methods).

>>>Answer: Thank you for your helpful suggestion. We agree that information regarding data sources is more appropriately placed in the Methods section. In accordance with your advice, we have removed the sentence referencing the NHANES database from the Introduction and integrated it into the first paragraph of the Methods section for improved structural coherence and readability.

Q3) METHODS

>>> Q3.1 Was this study approved by ethical committee? If so, I suggest to add the “approval number” or the “registration code”. All participants signed a written informed consensus before being enrolled the study.

>>>Answer: Thank you for raising this important point. Yes, the study was approved by the NCHS Research Ethics Review Board, and all participants provided written informed consent prior to participation. As suggested, we have now added the corresponding ethical approval information in the Methods section: “The Institutional Ethics Review Board of the National Centre for Health Statistics reviewed and approved the NHANES study protocols (Continuation of Protocol #2011-17), and all participants signed a written informed consensus before being enrolled the study.”

>>> Q3.2 When presenting the inclusion criteria, I suggest to briefly describe why you chose “20 years” as the cut-off for inclusion. On the one hand, due to the high prevalence of overweight in the American population, including such young patients might represent a strong point. On the other hand, including males older than 50 years might introduce a potential confounding factor, since in these subjects the testosterone reduction is at least in part cause by physiological reduction of testicular steroidogenesis over time, regardless from the presence of metabolic disease.

>>>Answer: Thank you for this thoughtful and well-balanced comment. We chose 20 years as the lower age limit for inclusion because NHANES provides complete biochemical and anthropometric data from this age onward, and early metabolic alterations, including obesity and insulin resistance, are increasingly prevalent among young adults in the United States. Including this age group allows us to explore the relationship between metabolic status and testosterone levels even in early adulthood, which we believe is a strength of the study. We fully acknowledge, however, that the inclusion of men over 50 years introduces potential confounding due to age-related physiological decline in testicular function and testosterone production, independent of metabolic status. To address this, we included age as a covariate in our statistical models and discussed this point as a potential limitation in the revised manuscript. These steps were taken to minimize confounding and improve the interpretability of our findings.

>>> Q3.3 When describing the technique for sample collection, I suggest to also add the time of collection (e.g., “samples were collected in the morning (before 10.00 a.m.) after at least eight hours of fasting”).

>>>Answer: Thank you for this valuable suggestion. We agree that specifying the time and fasting status of blood sample collection is important, especially for hormones such as testosterone that exhibit diurnal variation. Accordingly, we have updated the Methods section to include the following statement: “The blood samples were collected in the morning (before 10.00 a.m.) after at least eight hours of fasting.” This addition helps to clarify the pre-analytical conditions and supports the reliability of the hormone measurements used in our study.

>>> Q3.4 When describing all indexes considered, I suggest to point out the role of HOMA-index to make it explicit is role (I suppose it considered as a “gold-standard reference”)

>>>Answer: Thank you for this helpful comment. You are correct in noting that the HOMA-IR index was used as a reference indicator for insulin resistance in our study. It is widely recognized as a practical surrogate marker and is often considered the standard index in large-scale epidemiological studies. In accordance with your suggestion, we have clarified this point in the Methods section by explicitly stating the role of HOMA-IR as a reference for assessing insulin resistance. This addition helps contextualize the comparison between HOMA-IR and other metabolic indexes used in the study.

>>> Q3.5 Finally, when describing cut-off for diagnosing diabetes, please correct a small typing error ("blood glucose levels" is expressed in mg/dL instead of mmol/L: it should be 7.1 mmol/L and not 7.1 mg/dL).

>>>Answer: Thank you for pointing out this typographical error. We have corrected the unit in the revised manuscript: the diagnostic cut-off for diabetes is now correctly stated as 7.1 mmol/L (corresponding to 126 mg/dL). We appreciate your careful reading and attention to detail.

>>> Q3.6 When considering specifically the “statistical analysis section”:

>>> Q3.6.1 I suggest to add in the text whether and how continuous variables were testes for normal distribution (e.g., Shapiro Wilk test) and if/how were they transformed if requested (was testosterone serum level normally distributed or was it necessary to transform it before regression analysis?);

>>>Answer: Thank you for this important and technically relevant suggestion. In response, we have updated the Statistical Analysis section to include a description of how continuous variables were assessed for normality using the Shapiro–Wilk test. For variables that did not follow a normal distribution, including serum testosterone levels, log-transformation was applied prior to regression analysis to meet model assumptions. These steps were performed to ensure the validity and robustness of our statistical models, and the updated text now reflects this information: “Normality of continuous variables was assessed using the Shapiro–Wilk test. Variables with non-normal distribution, including serum testosterone levels, were log-transformed prior to regression analyses.”

>>> Q3.6.2 I suggest to rephrase when describing the types of regression used, in order to make it simpler for the reader to understand the logic underlying the statistic process (i.e., testosterone was considered both as continuous variable (➝ linear regression) and dichotomic variable (➝ logistic regression)).

>>>Answer: Thank you for this very helpful suggestion. In the revised version of the manuscript, we have rephrased the description of our regression models to clarify that serum testosterone was analyzed both as a continuous variable using linear regression and as a dichotomous variable (testosterone deficiency: yes/no) using logistic regression. This revised phrasing aims to enhance readability and to make the rationale behind our statistical approach more transparent to the reader: “The weighted regression models were employed to examine the associations between testosterone level and TD risk and various screening indexes (LAP, TyG, VAI, WC, BMI, TyG_WC, TyG_BMI, and TyG_WHtR). Specifically, linear regression models were used when serum total testosterone was treated as a continuous variable, while logistic regression models were applied when TD (TD, defined as TT < 300 ng/dL) was analyzed as a binary outcome.”

>>> Q3.6.3 When reading the manuscript and tables I guess that multivariate analysis was performed rather than simple ones: from this perspective I suggest to make it explicit into the text and adding into tables (i.e., table 2) the statistical parameters of every model (R, R2, p (test F) coefficient for linear regression and Likelihood ratio, R2, adjusted R2 and p for logistic one).

>>>Answer: We sincerely appreciate the reviewer’s careful reading and valuable suggestion. You are correct that multivariate regression analyses were performed to investigate the independent associations between screening indexes and both serum testosterone levels (linear regression) and testosterone deficiency (logistic regression). In response to your suggestion, we have now explicitly clarified this in the Statistical Analysis section and the relevant table captions. Regarding the inclusion of statistical parameters such as R, R², F-test p-values, and likelihood ratios: since all of our regression models were pe

---

## [Decision Letter · Decision Letter 1]

16 Apr 2025

Prediction of testosterone deficiency using different screening indexes in adult American men: an NHANES cross-sectional study

PONE-D-24-22453R1

Dear Dr. Feng,

We’re pleased to inform you that your manuscript has been judged scientifically suitable for publication and will be formally accepted for publication once it meets all outstanding technical requirements.

Kind regards,

Shaonong Dang, PhD

Academic Editor

PLOS ONE

Additional Editor Comments (optional):

Reviewers' comments:

Reviewer's Responses to Questions

**Comments to the Author**

1. If the authors have adequately addressed your comments raised in a previous round of review and you feel that this manuscript is now acceptable for publication, you may indicate that here to bypass the “Comments to the Author” section, enter your conflict of interest statement in the “Confidential to Editor” section, and submit your "Accept" recommendation.

Reviewer #1: All comments have been addressed

Reviewer #2: All comments have been addressed

2. Is the manuscript technically sound, and do the data support the conclusions?

Reviewer #1: Yes

Reviewer #2: Yes

3. Has the statistical analysis been performed appropriately and rigorously? 

Reviewer #1: Yes

Reviewer #2: Yes

4. Have the authors made all data underlying the findings in their manuscript fully available?

Reviewer #1: Yes

Reviewer #2: Yes

5. Is the manuscript presented in an intelligible fashion and written in standard English?

Reviewer #1: Yes

Reviewer #2: Yes

6. Review Comments to the Author

Reviewer #1: Dear authors,

I believe the authors have addressed all of my concerns, and the manuscript is now suitable for publication!

Reviewer #2: Authors have satisfactorily addressed all the concerns raised earlier. All the identified spelling mistakes have been corrected.

7. PLOS authors have the option to publish the peer review history of their article (what does this mean? ). If published, this will include your full peer review and any attached files.

**Do you want your identity to be public for this peer review?** For information about this choice, including consent withdrawal, please see our Privacy Policy .

Reviewer #1: No

Reviewer #2: **Yes: ** Professor Sikiru Adetona OLURODE

---

## [Editor Report · Acceptance letter]

PONE-D-24-22453R1

PLOS ONE

Dear Dr. Feng,

I'm pleased to inform you that your manuscript has been deemed suitable for publication in PLOS ONE. Congratulations! Your manuscript is now being handed over to our production team.

Kind regards,

on behalf of

Dr. Shaonong Dang

Academic Editor

PLOS ONE